# Silencing of the Ortholog of *DEFECTIVE IN ANTHER DEHISCENCE 1* Gene in the Woody Perennial *Jatropha curcas* Alters Flower and Fruit Development

**DOI:** 10.3390/ijms21238923

**Published:** 2020-11-24

**Authors:** Chuan-Jia Xu, Mei-Li Zhao, Mao-Sheng Chen, Zeng-Fu Xu

**Affiliations:** 1CAS Key Laboratory of Tropical Plant Resources and Sustainable Use, Xishuangbanna Tropical Botanical Garden, Innovative Academy for Seed Design, Chinese Academy of Sciences, Menglun, Mengla 666303, Yunnan, China; xuchuanjia@xtbg.ac.cn (C.-J.X.); zhaomeili@xtbg.ac.cn (M.-L.Z.); 2Center of Economic Botany, Core Botanical Gardens, Chinese Academy of Sciences, Menglun, Mengla 666303, Yunnan, China; 3College of Life Sciences, University of Chinese Academy of Sciences, Beijing 100049, China

**Keywords:** DAD1, flower and fruit development, physic nut, jasmonic acid, RNA interference

## Abstract

DEFECTIVE IN ANTHER DEHISCENCE 1 (DAD1), a phospholipase A1, utilizes galactolipids (18:3) to generate α-linolenic acid (ALA) in the initial step of jasmonic acid (JA) biosynthesis in *Arabidopsis thaliana*. In this study, we isolated the *JcDAD1* gene, an ortholog of *Arabidopsis DAD1* in *Jatropha curcas*, and found that it is mainly expressed in the stems, roots, and male flowers of *Jatropha*. *JcDAD1*-RNAi transgenic plants with low endogenous jasmonate levels in inflorescences exhibited more and larger flowers, as well as a few abortive female flowers, although anther and pollen development were normal. In addition, fruit number was increased and the seed size, weight, and oil contents were reduced in the transgenic *Jatropha* plants. These results indicate that *JcDAD1* regulates the development of flowers and fruits through the JA biosynthesis pathway, but does not alter androecium development in *Jatropha*. These findings strengthen our understanding of the roles of JA and *DAD1* in the regulation of floral development in woody perennial plants.

## 1. Introduction

Jasmonic acid (JA) is one of several major phytohormones that plays a pivotal role in modulating plant growth and development, as well as in responding to various abiotic and biotic stresses [1,2,3,4]. JA is synthesized from α-linolenic acid (ALA) via the octadecanoid pathway [5,6]. The initial step of JA biosynthesis is the release of ALA from fatty acids in *Arabidopsis* chloroplasts, which is performed by two types of enzymes, fatty acid desaturases (FADs) and phospholipase A1 (PLA1) [7]. FADs convert diunsaturated fatty acids (18:2) to triunsaturated fatty acids (18:3), whereas PLA_1_ proteins, including DONGLE (DGL), DEFECTIVE IN ANTHER DEHISCENCE 1 (DAD1), and other lipases, hydrolyze the triunsaturated fatty acids in glycerides and phospholipids to produce free ALA [7]. Three sequential steps are catalyzed by 13-lipoxygenases (13-LOXs), allene oxide synthase (AOS), and allene oxide cyclase (AOC): oxygenation of ALA to 13-hydroperoxylinoleic acid (13-HPOT), dehydrogenation of 13-HPOT to the unstable compound 12,13-epoxyoctadecatrienoic acid (12,13-EOT), and cyclization of 12,13-EOT to (9S,13S)-12-oxo-phytodienoic acid (OPDA), respectively [8,9,10,11,12,13]. OPDA is synthesized in chloroplasts, transported to the peroxisomes by the transport protein peroxisomal ABC transporter 1 (PXA1) [14] and then converted to 3-oxo-2-(2′(Z)-pentenyl)-cyclopentane-1-octanoic acid (OPC-8:0) by 12-oxo-phytodienoic acid reductase (OPR) [12]. Subsequently, OPC-8:0 in peroxisomes is converted to JA by three cycles of β-oxidation [15,16,17]. Eventually, JA is catalyzed via jasmonic acid carboxyl methyltransferase (JMT) to methyl jasmonate (MeJA) or by jasmonic acid-amino acid synthetase to jasmonic acid-amino acids, such as jasmonic acid-isoleucine (JA-Ile) [18].

The *DAD1* gene encodes a chloroplastic glycerolipid lipase that belongs to the PLA_1_ family and acts in the initial step of JA biosynthesis [19,20,21]. In *Arabidopsis*, the *defective in anther dehiscence 1* (*dad1*) mutant displays a male sterility phenotype resulting from decreased endogenous JA levels, deficient filament elongation, nonviable pollen, and abnormal anther dehiscence, all of which can be rescued by exogenous treatment with linolenic acid or methyl jasmonate [22]. *Arabidopsis* plants with mutations in genes encoding key enzymes in JA biosynthesis (e.g., *fad3−2 fad7−2 fad8*, *lox3 lox4*, *aos*) exhibit similar phenotypes [23,24,25]. Transgenic *Arabidopsis* overexpressing *DAD1* shows various pale color phenotypes because of the destruction of chloroplasts caused by the excessive accumulation of DAD1 protein in these organelles [22,26]. Antisense suppression of *DAD1* in *Brassica rapa* induces male sterility and delays or inhibits flower opening [27]. In rice (*Oryza sativa*), mutation of *EXTRA GLUME 1* (*EG1*), which is a homolog of *Arabidopsis DAD1*, causes a decrease in the endogenous JA level, resulting in a changed spikelet morphology that includes altered floral organ number, increased glume-like structures and defective floral meristem determinacy [28]. In maize (*Zea mays*), *Tasselseed1* (*TS1*) encodes a plastid-targeted 13-LOX that acts in JA biosynthesis [29,30], and two oxophytodienoate reductase genes, *OPR7* and *OPR8*, also participate in JA production [31]. Mutations of either *TS1* or *OPR7*/*OPR8* lead to the conversion of tassel inflorescence from staminate to pistillate [28,30,31,32]. These results indicate that JA plays diverse roles in flower development and sex determination. The functions of *DAD1* have been successively revealed in herbaceous plants [19,33,34,35,36,37] but are still unknown in woody perennial plants.

*Jatropha curcas* L., a woody perennial plant, is considered an important biofuel plant with economic value because it contains a high content of seed oil (30−40%) that can be processed into biodiesel and aviation oil [38,39,40,41]. Typically, *Jatropha* is monoecious, with separate female and male flowers on the same inflorescence [42,43]. In a previous study, we speculated that *JcDAD1*, an ortholog of *Arabidopsis thaliana DAD1*, might participate in the abortion of male flowers during the transition from monoecy to gynoecy in *Jatropha* [44]. In this study, we repressed expression of *JcDAD1* by gene silencing to investigate its function in the regulation of flower and fruit development in *Jatropha.*

## 2. Results

### 2.1. Characterization of the JcDAD1 Gene

We obtained a *JcDAD1* cDNA library (GenBank accession no. 105643375) from the *Jatropha* transcriptome [44]. The lengths of the *JcDAD1* genomic sequence and coding sequence (CDS) are 1537 bp and 1323 bp, respectively (http://jcdb.xtbg.ac.cn/). The *JcDAD1* gene, which encodes 440 amino acids, is located on the fifth chromosome of *Jatropha* [45]. Similar to *Arabidopsis* and rice, *JcDAD1* possesses only one exon accompanied by no introns (Figure 1A), indicating that *DAD1* is evolutionarily conserved among these plants. Phylogenetic analysis showed that JcDAD1 has a close relationship with HbDAD1 and RcDAD1 (Figure 1B). In addition, the JcDAD1 protein containing a phospholipase A1 domain belongs to a member of the alpha/beta-hydrolase superfamily of proteins, which is similar to that of *Arabidopsis* [46].

The tissue-specific expression analysis showed that *JcDAD1* is primarily expressed in the stems, roots, and male flowers but has low expression levels in other tissues of *Jatropha* (Figure 2). In *Arabidopsis*, expression of *DAD1* is highly restricted to the filaments of stamens, which is consistent with the function of JA in promoting water transport by synchronizing anther dehiscence, pollen maturation, and flower opening [22]. In rice, expression of the *EG1* gene (an ortholog of *DAD1*) is high in inflorescence primordia, but weak in developing floral primordia, which is in accordance with its role in early flower development [28,47]. However, the transcript of tomato *LeLID1* (a homolog of *Arabidopsis DAD1*) is hardly detected in reproductive organs (buds, flowers, or fruits) but strongly expressed in germinating seedlings, where the encoded protein functions as a TAG lipase [48]. These results show that *DAD1* genes have various expression patterns in different species, indicating that they may have different functions.

### 2.2. JcDAD1 Gene Silencing Increased Inflorescence Branching, Flower Number, and Flower Size

To investigate the functionality of the *JcDAD1* gene, we transformed the *JcDAD1*-RNAi construct into *Jatropha* plants and identified 16 independent *JcDAD1*-RNAi transgenic lines (T1 generation), in which the transcript of *JcDAD1* is repressed in inflorescence buds (Figure 3). The order of inflorescence branches was increased to the fifth-order branches in the transgenic inflorescence, whereas the wild type (WT) inflorescence had only fourth-order branches (Figure 4). Compared with the WT plants, notable increases in the female flower number (approximately three-fold) and male flower number (three- to five-fold) per inflorescence were observed in the transgenic *Jatropha* plants (Figure 5A−F), which is similar to the phenotype of the *Arabidopsis dad1* mutant [22]. The flower size of the transgenic plants was conspicuously larger than that of the WT plants (Figure 6A−L), and the average diameters of the transgenic female and male flowers increased by 2−4 mm (Figure 6M). These results show that *JcDAD1* participates in flower development and promotes flower production and floral organ growth in *Jatropha*.

### 2.3. JcDAD1 Gene Silencing Caused the Abortion of Some Female Flowers but Did Not Affect Anther Dehiscence

At the late stage of flower development (in the inflorescences of 21–30 days after emergence), a portion of flowers were abortive and most were female flowers in the transgenic inflorescences, while flower development was normal in the WT inflorescences (Figure 7A−L). Abortion might be caused by a deficiency of the nutrient supply. In *Arabidopsis*, the *dad1* mutant displays a defect in anther dehiscence [22], although obvious phenotypic changes in anthers and pollens were not observed in the transgenic *Jatropha* (Figure 8). These results suggest that DAD1 may play different roles in the regulation of anther dehiscence and pollen development in *Arabidopsis* and *Jatropha*.

### 2.4. JcDAD1 Gene Silencing Affected Jatropha Yield Traits

Compared with that of WT *Jatropha*, the fruit number per infructescence in the transgenic plants was increased by 2–3-fold (Figure 9A−E). The transgenic *Jatropha* plants produced smaller fruit and seeds in length and width (Figure 10A−G), and had a lighter ten-seed weight (Figure 10H), and lower seed oil content (Figure 10I). The results indicate that *JcDAD1* regulates fruit development and therefore affects the traits of seeds in *Jatropha*.

### 2.5. JcDAD1 Gene Silencing Reduced Endogenous JA and JA-Ile Contents in Jatropha Inflorescences

To determine whether the endogenous jasmonate contents in the transgenic *Jatropha* plants were affected by *JcDAD1*, two types of jasmonate (JA and the bioactive form JA-Ile) were measured in developing inflorescences (15−20 days after emergence) from the WT and transgenic *Jatropha* plants. Based on the results, the concentrations of both JA and JA-Ile were significantly decreased in the transgenic *Jatropha* inflorescences (Figure 11), indicating that *JcDAD1* is a key positive regulator of JA biosynthesis.

## 3. Discussion

JA, a pivotal phytohormone, plays diversified functions in inflorescence and flower development [49]. The initial step of JA biosynthesis is catalyzed by DAD1 [19,20,21,50]. The *Arabidopsis dad1* mutant generally displayed male sterility and abnormal anther dehiscence that can be rescued by exogenous methyl jasmonate (MeJA) treatment [22]. Similarly, antisense inhibition of *BrDAD1* also resulted in male sterility in *Brassica rapa* [27]. Compared with the male sterile phenotypes in *Arabidopsis* and *Brassica*, *EG1*, an ortholog of *Arabidopsis DAD1*, controlled both empty glume fate and spikelet development in rice [22,27,28,47,49]. In this study, *JcDAD1* silencing caused a decrease of endogenous JA levels and an increase in flower number and flower size in the *JcDAD1*-RNAi transgenic inflorescences (Figure 3, Figure 5, Figure 6 and Figure 11), suggesting that *JcDAD1* regulates flower development by controlling JA levels in *Jatropha* inflorescences. However, the *JcDAD1*-RNAi transgenic plants did not exhibit obviously abnormal androecium (Figure 8), which is not consistent with that in *Arabidopsis*, thus implying that JA could play different roles in regulating androecium development in *Jatropha* and *Arabidopsis*. Furthermore, the phenotype of male flowers showed increased size but normal fertility in the transgenic plants, which is inconsistent with the speculation that *JcDAD1* might contribute to male abortion in gynoecious *Jatropha* plants [44]. It is possible that *JcDAD1* may function in adjusting the balance between reproductive development and stress response via the JA synthetic pathway because *DAD1* can simultaneously act in flower development and wound defense and inhibit pathogen infection [7,22]. In general, cell size or cell number, which are mainly controlled by the genes involved in the biosynthesis or signal transduction of auxin, ethylene, cytokinin, and brassinosteroid, determine flower size in the plant kingdom [51,52]. An increase in flower size in *JcDAD1*-RNAi transgenic *Jatropha* plants (Figure 6) suggests that JA might participate in the regulation of flower size in woody perennial plants. In the transgenic *Jatropha* plants, some female flowers and a few male flowers were abortive at the early stage of inflorescence development (Figure 7), which is likely caused by insufficient nutrient supply due to the generation of more flowers in a single inflorescence. In summary, this study suggests that *DAD1* genes are involved in the JA biosynthesis pathway and play diverse roles in regulating flower development among different species.

To investigate whether abnormal flower phenotypes of the transgenic plants can be rescued by exogenous jasmonate treatment, JA (2.5 or 5 mM) and MeJA (200 or 400 μM) solutions were sprayed onto the emerging transgenic inflorescence buds in the field. Unfortunately, whether JA/MeJA treatment could recover flowers in the transgenic inflorescences to normal size or normal flower number could not be clarified. Given the lack of control of environmental cues in the field trials, an optimized scheme will be designed for conducting experiments in future studies. Treatment with exogenous gibberellic acid (GA) was found to increase the number of female flowers in *Jatropha* [53,54,55], similar to the phenotypes of the *JcDAD1*-RNAi transgenic plants (Figure 5E). Moreover, both exogenous benzyladenine (BA) treatment and flower-specific overproduction of endogenous cytokinins (CKs) promoted the total flower number and female flower number in *Jatropha* [38,56], which again resembled the phenotypes of the *JcDAD1*-RNAi transgenic plants (Figure 5E,F). These results indicate that JA may act together with GA and/or CK to regulate flower development in *Jatropha*. However, the underlying molecular mechanism remains unclear and needs further investigation in the future.

## 4. Materials and Methods

### 4.1. Plant Materials

Two-year-old WT and *JcDAD1*-RNAi transgenic *Jatropha* plants were cultivated on the hillside land of an experimental field located in the Xishuangbanna Tropical Botanical Garden (XTBG; 21°54′ N, 101°46′ E; 580 m in altitude) of the Chinese Academy of Sciences, Menglun County, Yunnan Province, Southwest China [38]. Inflorescences from the WT and transgenic T1 generation plants were collected for morphologic observation. Roots, stems, young leaves, mature leaves, developing inflorescences (15−20 days after emergence), inflorescence buds, female flowers, male flowers, pericarps, and seeds were harvested from WT or transgenic plants, immediately frozen in liquid nitrogen and then stored at −80 °C for quantitative reverse transcriptase-polymerase chain reaction (qRT-PCR) analysis.

### 4.2. Sequence Alignment and Phylogenetic Analysis

The gene structures of *DAD1* in different species were analyzed by the Gene Structure Display Server (GSDS2.0, Center for Bioinformatics, Peking University, China) with default settings [57]. The deduced DAD1 amino acid sequences for phylogenetic analysis from NCBI databases were used. A phylogenetic tree was constructed with the neighbor-joining statistical method, the Poisson model, and the bootstrap method applied in 1000 replications via MEGA (version 7.0) software (http://www.megasoftware.net/).

### 4.3. Isolation of JcDAD1 cDNA

Total RNA was extracted from *Jatropha* leaves using the pBIOZOL Plant Total RNA Extraction Reagent (BioFlux, Hangzhou, China) following the manufacturer’s instructions. Then, first-strand cDNA was synthesized with 1.0 μg of the extracted RNA using the TAKARA PrimeScript^TM^ RT Reagent Kit (TAKARA, Dalian, China). A 153 bp fragment (TCCGTCAATCAGATGGAGATACGTGCTTAGCTCGTGACATGTGGCCACGTCTCTTTTGTTTAGATACGGTGACTGTTTGCTACTGAGCCTAAGCTCTTGCCCCACCTCAGCATAAACCCACTGCGTGCTCTCTACTCGCTTCTGTAACCAACT) was PCR-amplified from the synthesized cDNA using primers carrying appropriate restriction enzyme cutting sites (all primers used in this study are listed in Appendix A). The PCR products were subsequently cloned into a pEASY-Blunt Zero cloning vector (TransGen Biotech, Beijing, China) with the appropriate restriction enzymes and sequenced.

### 4.4. RNAi Silencing Vector Construction and Transformation

To construct the *JcDAD1*-RNAi expression vector, the sense and antisense fragments of *JcDAD1* were cloned into the pJL10 binary vector [58] in opposing orientations on either side of a pdk intron to produce an invert repeat driven by the 35S promoter. The expression vector was transformed into *Jatropha* with *Agrobacterium* strain EHA105 as described previously [59]. The positive transgenic plants were confirmed by PCR and qRT-PCR.

### 4.5. Quantitative RT-PCR (qRT-PCR)

qRT-PCR was performed using SYBR^®^ Premix Ex Taq^TM^ II (TAKARA, Dalian, China) on a Roche 480 Real-Time PCR Detection System (Roche Diagnostics, Mannheim, Germany). At least two biological replicates and three technical replicates for all samples were applied in the qRT-PCR analysis. We used the 2^−ΔΔ*C*T^ method described by Livak and Schmittgen [60] to analyze the data. The *JcActin* gene was used to normalize the transcript levels of specific genes of *Jatropha*.

### 4.6. Phenotypic Analysis of Flowers

The phenotypes of heterozygous (T1 generation) transgenic *Jatropha* plants were analyzed using a 3D Super Depth digital microscope (Smart Zoom 5, Carl Zeiss, Germany,).

### 4.7. Quantitation of the JA and JA-Ile Contents

The 15- to 20-day-old inflorescences of the *JcDAD1*-RNAi transgenic and WT plants were harvested, frozen rapidly in liquid nitrogen and stored at −80 °C for measuring the JA and JA-Ile contents. The measurement method was described previously [61].

## 5. Conclusions

Silencing of *JcDAD1* reduced endogenous jasmonate contents in inflorescences, increased the size and number of flowers, and caused the abortion of a few female flowers in *Jatropha*. Furthermore, the *JcDAD1*-RNAi transgenic *Jatropha* plants displayed increases in fruit number and decreases in seed size, weight and oil contents. However, compared with *Arabidopsis*, anther and pollen development in the *JcDAD1*-RNAi transgenic *Jatropha* plants was normal. These results indicate that *JcDAD1* participates in JA biosynthesis and acts in regulating flower and fruit development in *Jatropha*. *JcDAD1* is also highly expressed in the stems and roots, implying that *JcDAD1* not only plays important roles in reproductive growth, but may also have undiscovered roles in vegetative growth. Since synergy may occur among JA, GA, and/or CK in regulating flower development of *Jatropha*, interactions of JA and other phytohormones to control flower development in *Jatropha* will be investigated in further work.

## Figures and Tables

**Figure 1 ijms-21-08923-f001:**
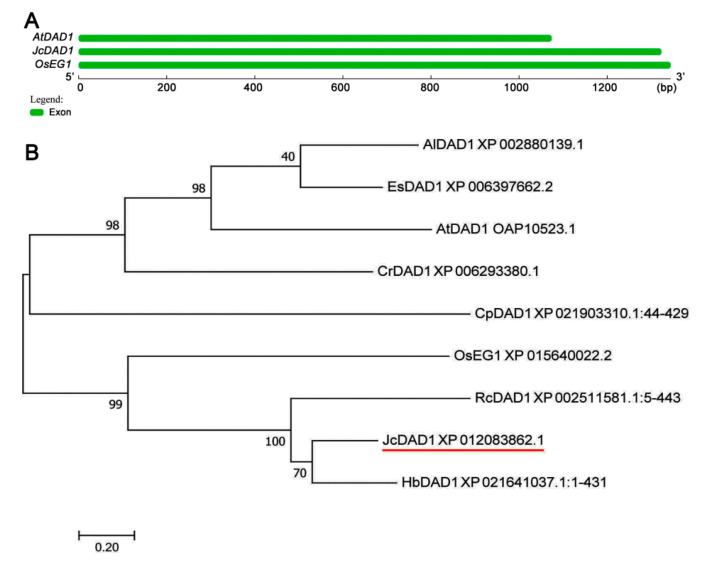
Structural and phylogenetic analysis of *DAD1* genes among *Jatropha* and other species. (**A**) Genomic organization of *DAD1*. (**B**) Phylogenetic analysis of deduced DAD1 proteins. *EG1*, a rice ortholog of *Arabidopsis DAD1*. Al, *Arabidopsis lyrata*; At, *Arabidopsis thaliana*; Cp, *Carica papaya*; Cr, *Capsella rubella*; Es, *Eutrema salsugineum*; Hb, *Hevea brasiliensis*; Jc, *Jatropha curcas*; Os, *Oryza sativa*; Rc, *Ricinus communis*. The GenBank accession numbers in the phylogenetic tree are listed in Appendix A.

**Figure 2 ijms-21-08923-f002:**
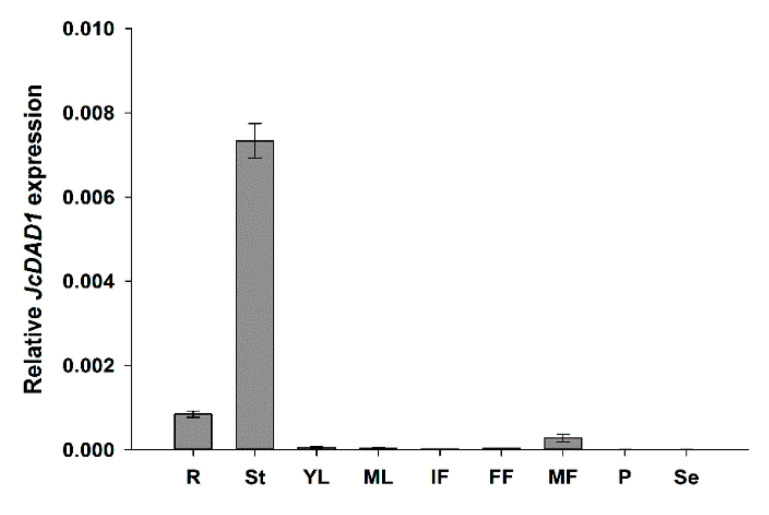
Tissue-specific expression analysis of *JcDAD1* in different tissues of adult *Jatropha* plants. Three biological replicates and three technical replicates were prepared for qRT-PCR. *JcActin* was used as the internal reference. Error bars represent standard errors (*n* = 3). FF, female flowers; IF, inflorescences; MF, male flowers; ML, mature leaves; P, pericarps; R, roots; Se, seeds; St, stems; YL, young leaves.

**Figure 3 ijms-21-08923-f003:**
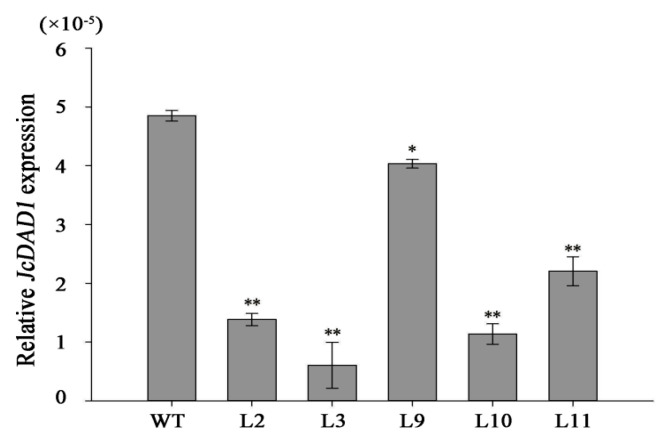
Relative expression levels of *JcDAD1* in inflorescence buds of the wild type (WT) plants and *JcDAD1*-RNAi transgenic *Jatropha* lines (L2, L3, L9, L10, and L12). Two biological replicates and three technical replicates were prepared for qRT-PCR assays. *JcActin* was used as the internal reference. Error bars represent standard errors (*n* = 2). Student’s *t*-test was performed to analyze significant differences. ** Extremely significant difference (*p* < 0.01). * Significant difference (*p* < 0.05).

**Figure 4 ijms-21-08923-f004:**
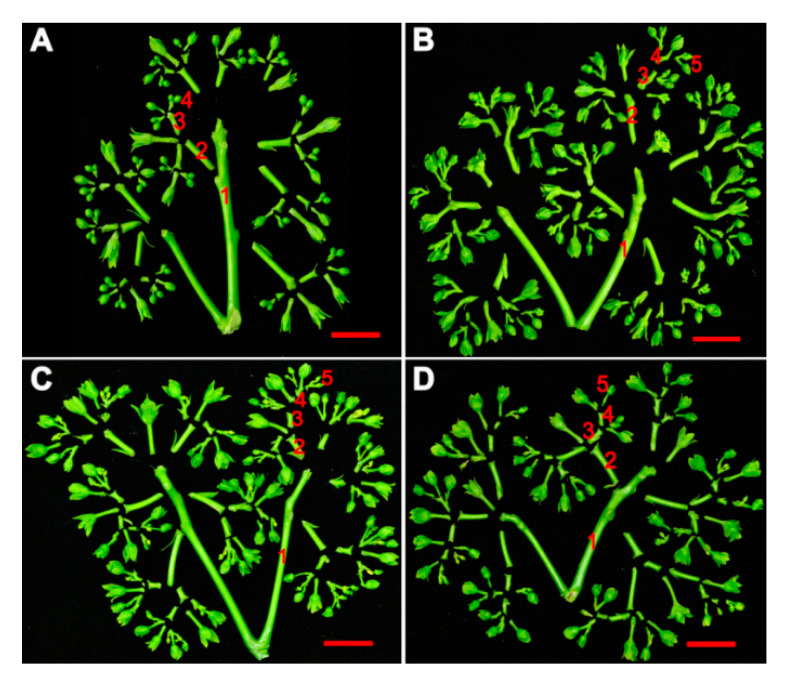
Comparison of inflorescence branching in the *JcDAD1*-RNAi transgenic plants with that in the wild type (WT) plants. The graph depicts inflorescence branching in WT plants (**A**) and the transgenic lines L2 (**B**), L3 (**C**), and L10 (**D**). The numbers (1–5) represent different orders of branching. Bars = 2.0 cm.

**Figure 5 ijms-21-08923-f005:**
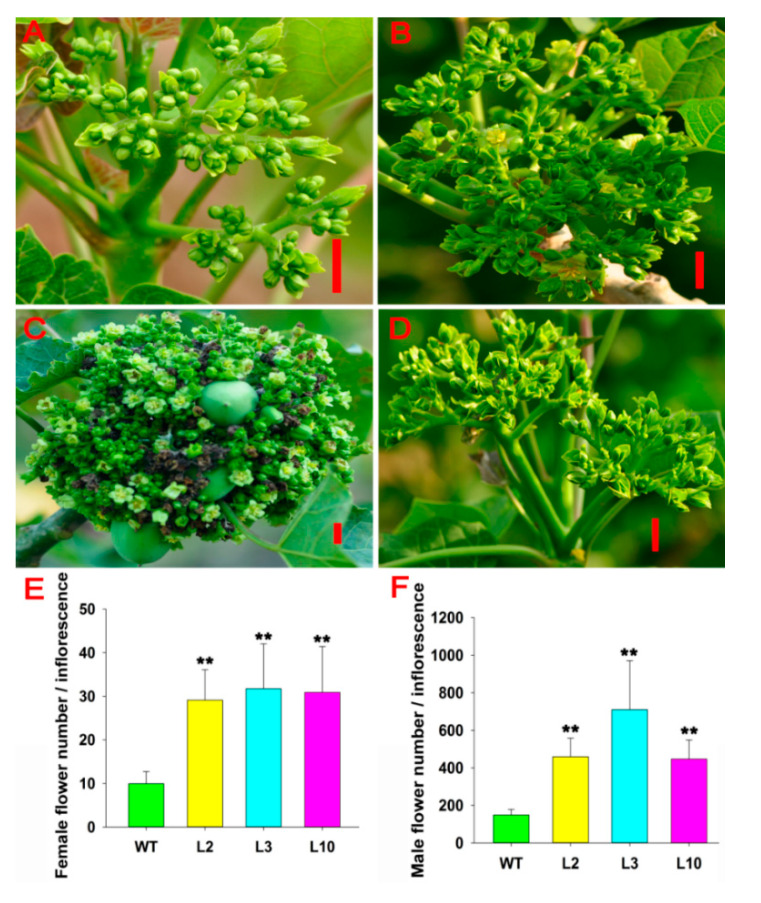
Number of flowers per inflorescence was increased in the *JcDAD1*-RNAi transgenic plants. (**A−D**) Inflorescences of wild-type (WT) plants (**A**), and inflorescences of transgenic lines L2 (**B**), L3 (**C**), and L10 (**D**); bars = 1.0 cm. (**E**,**F**) Comparison of female (**E**) and male (**F**) flower number per inflorescence in the transgenic lines (L2, L3 and L10) with those in the WT plants, *n* ≥ 8. Error bars represent the standard deviations. Student’s *t*-test was performed to analyze significant differences. ** Extremely significant difference (*p* < 0.01).

**Figure 6 ijms-21-08923-f006:**
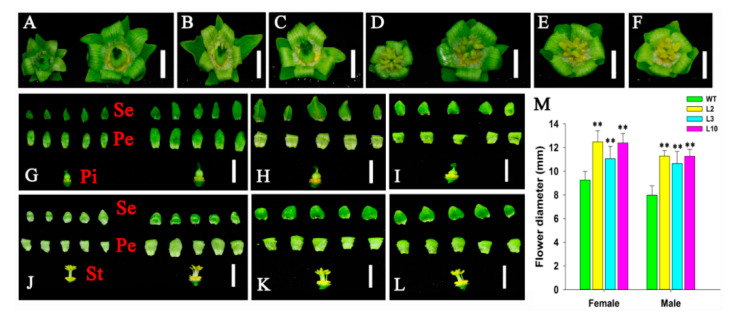
Comparison of the flower morphology between the wild-type (WT) and *JcDAD1*-RNAi transgenic plants. (**A**) Female flowers in the WT (left) and transgenic line L2 (right). (**B**,**C**) Female flowers of the transgenic lines L3 and L10, respectively. (**D**) Male flowers of the WT (left) and transgenic line L2 (right) *Jatropha* plants. (**E**,**F**) Male flowers of the transgenic lines L3 and L10, respectively. (**G**) Anatomy of female flowers of the WT (left) and transgenic line L2 (right). (**H**) and (**I**) Anatomy of the female flowers of the transgenic lines L3 and L10, respectively. (**J**) Anatomy of the male flowers of the WT (left) and transgenic line L2 (right). (**K**,**L**) Anatomy of the male flowers of the transgenic lines L3 and L10, respectively. Bars = 0.5 cm shown in (**A**−**F**); bars = 1.0 cm shown in (**G**−**L**). (**M**) Statistics of female and male flower size in the WT and transgenic *JcDAD1*-RNAi plants, *n* ≥ 31. Pe, petal; Pi, pistil; Se, sepal; St, stamen. Error bars represent the standard deviations. Student’s *t*-test was performed to analyze significant differences. ** Extremely significant difference (*p* < 0.01).

**Figure 7 ijms-21-08923-f007:**
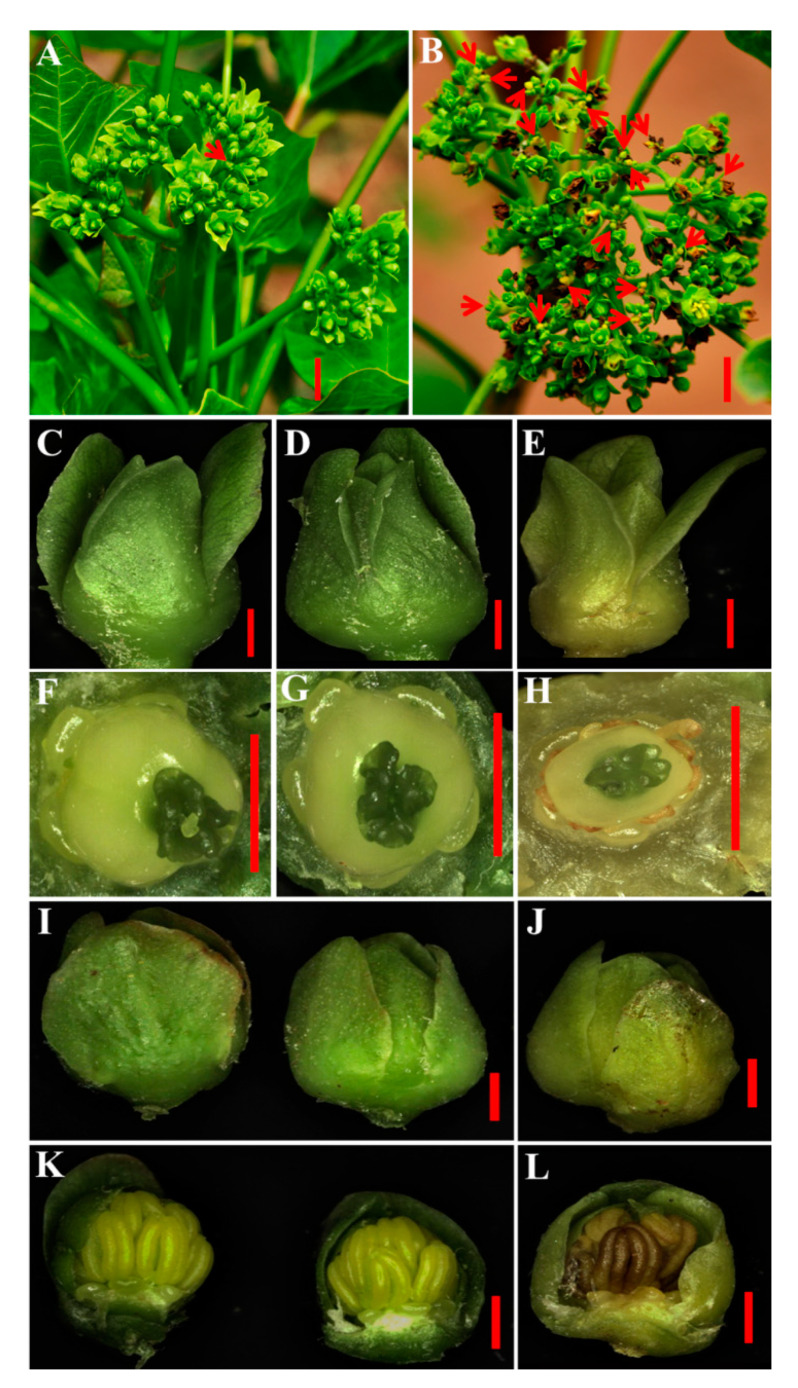
Abortion of florets in transgenic *Jatropha* plants. (**A**,**B**) Inflorescences of the wild type (WT) and transgenic line L10, respectively. The red arrow refers to the normal floret in (**A**) or the abnormal florets in (**B**). (**C**,**D**) Normal female flowers of WT and transgenic L10 *Jatropha*, respectively. (**E**) Abortive female flowers of transgenic L10 *Jatropha*. (**F**−**H**) Anatomy of female flowers from (**C**−**E**). (**I**) Normal male flowers in the WT (left) and transgenic *Jatropha* L10 (right). (**J**) Abortive male flowers of the transgenic *Jatropha* L10. (**K**,**L**) Anatomy of the male flowers from (**I**,**J**), respectively. Bar**s** = 1.0 cm shown in (**A**,**B**); bars = 1.0 mm shown in (**C−L**).

**Figure 8 ijms-21-08923-f008:**
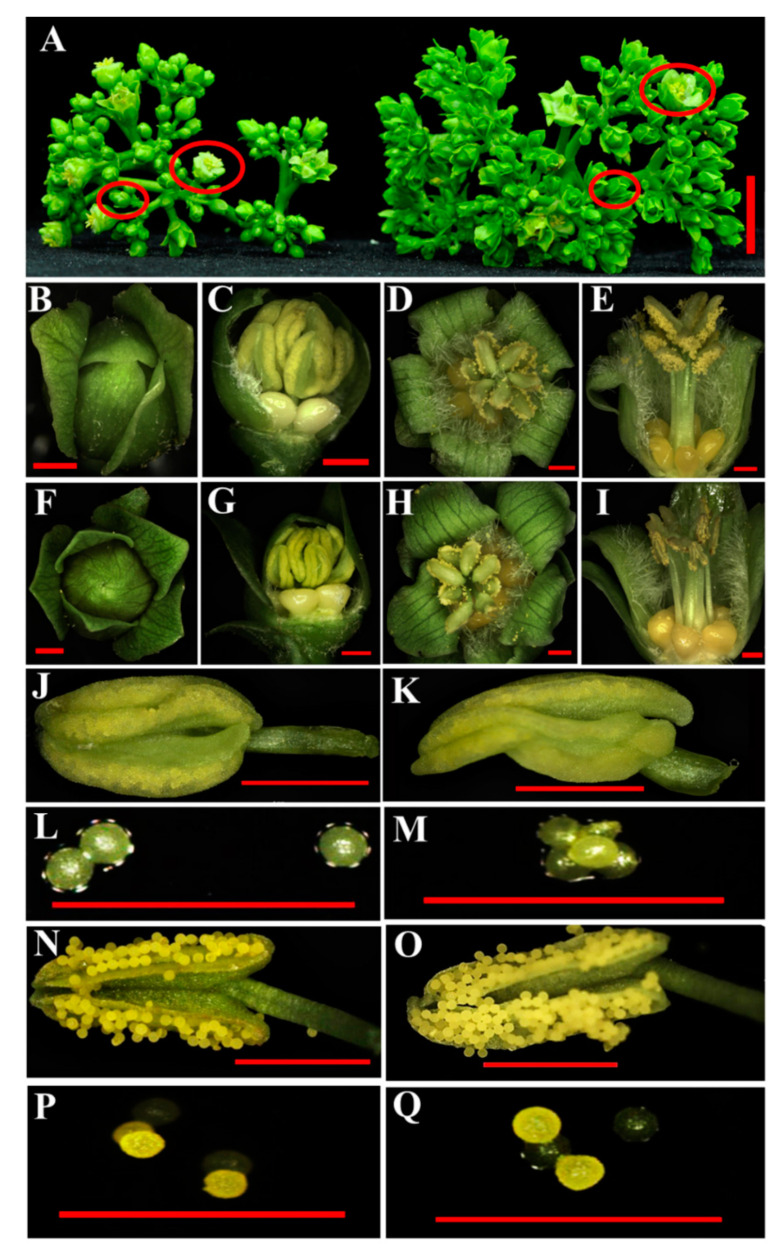
Morphology of the anthers and pollen grains in the *JcDAD1*-RNAi transgenic and wild type (WT) plants. (**A**) Inflorescences of the WT (left) and transgenic line L2 (right). Small circles shown in (**A**) represent the flowers shown in (**B**,**F**); and large circles shown in (**A**) represent the flowers shown in (**D**,**H**). (**B**), (**C**), (**F**) and (**G**) Unopened flowers. (**D**), (**E**), (**H**) and (**I**) Opened flowers. (**C**), (**E**), (**G**) and (**I**) Anatomy of the flowers shown in (**B**), (**D**), (**F**) and (**H**), respectively. (**B**−**E**) Male flowers from the WT plants. (**F**−**I**) Male flowers from the transgenic L2 plants. (**J**,**K**) Indehiscent anthers. (**N**) and (**O**) Dehiscent anthers. (**L**,**M**) Pollen grains from the anthers shown in (**J**,**K**), respectively. (**N**,**O**) Pollen grains from the anthers shown in (**P**,**Q**), respectively. (**J**), (**L**), (**N**) and (**P**) Male flowers of the WT plants. (**K**), (**M**), (**O**), and (**Q**) Male flowers of the transgenic L2 *Jatropha* plants. Bar = 2.0 cm shown in (**A**); bars = 1.0 mm shown in (**B**−**K**), (**N**,**O**); bars = 500.0 μm shown in (**L**), (**M**), (**P**) and (**Q**).

**Figure 9 ijms-21-08923-f009:**
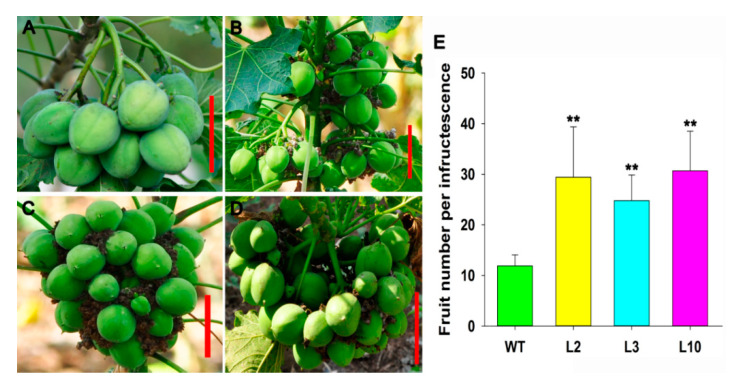
Fruit number per infructescence was increased in the *JcDAD1*-RNAi transgenic plants. (**A**−**D**) Infructescences of the wild type (WT) and transgenic lines L2, L3, and L10, respectively; scale bars = 5.0 cm. (**E**) Comparison of the fruit number per infructescence between the WT and transgenic lines, *n* ≥ 5. Error bars represent the standard deviations. Student’s *t*-test was performed to analyze significant differences. ** Extremely significant difference (*p* < 0.01).

**Figure 10 ijms-21-08923-f010:**
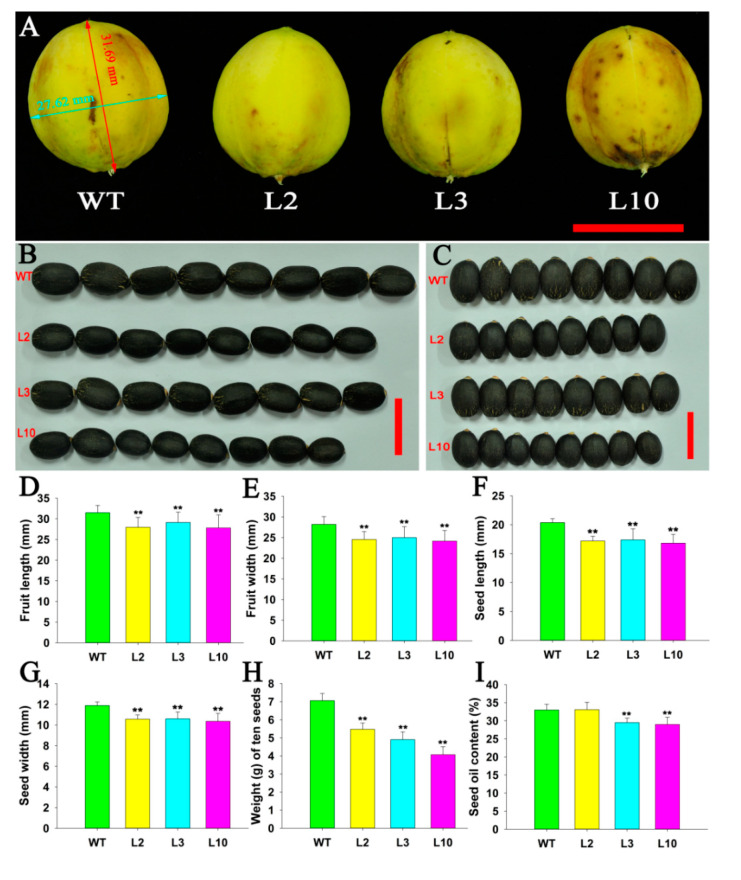
Decrease in the fruit size, seed size, seed weight, and seed oil content in the *JcDAD1*-RNAi transgenic plants. (**A**) Fruits from the wild type (WT) and transgenic *Jatropha* plants. The red line with the double-headed arrow represents the fruit length (31.69 mm), and the cyan line represents the fruit width (27.62 mm). (**B**,**C**) Seed length and width in the WT and transgenic *Jatropha* plants. (**D**,**E**) Fruit length and width in the WT and transgenic *Jatropha* plants, *n* ≥ 34. (**F**,**G**) Seed length and width in the WT and transgenic *Jatropha* plants, *n* ≥ 120. (**H**) Ten-seed weight in the WT and transgenic *Jatropha* plants, *n* ≥ 14. (**I**) Seed oil content in the WT and transgenic *Jatropha* plants, *n* ≥ 14. Scale bars = 2.0 cm. L2, L3 and L10 represent different transgenic lines. Error bars represent standard deviations. Student’s *t*-test was performed to analyze significant differences. ** Extremely significant difference (*p* < 0.01).

**Figure 11 ijms-21-08923-f011:**
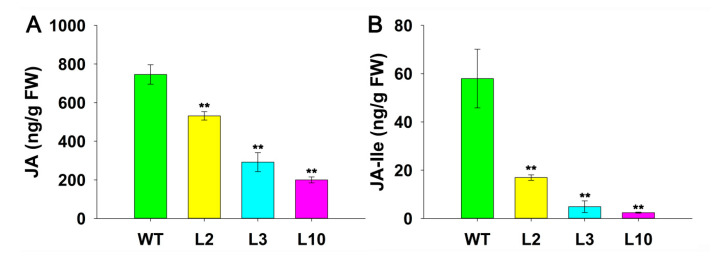
Contents of JA (**A**) and JA-Ile (**B**) in early developing inflorescences from the wild type (WT) and the transgenic *Jatropha* plants. FW, fresh weight; JA, jasmonic acid; JA-Ile, jasmonic acid-isoleucine. Error bars represent the standard deviations (*n* ≥ 3). Student’s *t*-test was performed to analyze significant differences. ** Extremely significant difference (*p* < 0.01).

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
