# Peer review of "Silencing of the Ortholog of DEFECTIVE IN ANTHER DEHISCENCE 1 Gene in the Woody Perennial Jatropha curcas Alters Flower and Fruit Development"

_ijms, 2020, doi:10.3390/ijms21238923_

Round 1

Reviewer 1 Report

The manuscript deserve to be published for the data included, but it needs a deep English / style revision; in fact, some of the errors / misspellings are important, for example Deffective with two "f"instead of the correct “Defective”, a very technical term, is incorrect in the MS title.

Moreover, other important details:

  1. Authors must modify the Figures 4-10 to have bars of the same size (for comparable organs) to permit a correct comparison otherwise, e.g. in Figure 4 which is the largest inflorescence, the A or the B?
  2. It would be appropriate to indicate the starting and ending base of the fragment of 153 bp employed for cloning in the RNAi expression vector;
  3. In Figures 2, 5 and 11, it would be better to avoid the interruption of the Y axis, perhaps separating into two figures in the case of male and female inflorescences for Fig. 5 (the total has no meaning given the number of female inflorescences), and separating also JA and JA-Ile for Fig. 11.

Author Response

Point 1: The manuscript deserve to be published for the data included, but it needs a deep English / style revision; in fact, some of the errors / misspellings are important, for example Deffective with two "f"instead of the correct “Defective”, a very technical term, is incorrect in the MS title.

Response: Thank you very much for your suggestion. We revised the English language in detail as well as the style of the manuscript, and the revisions are highlighted in red text.

Point 2: Authors must modify the Figures 4-10 to have bars of the same size (for comparable organs) to permit a correct comparison otherwise, e.g. in Figure 4 which is the largest inflorescence, the A or the B?

Response: Thank you very much for your suggestion. We checked and carefully modified the size of the bars shown in Figures 4-10 so that the size of the organs could be easily compared. In addition, some errors were amended in this manuscript. For example, the scale bars = 1.0 cm but not 2.0 cm as shown in Figure 5, which was revised.

Point 3: It would be appropriate to indicate the starting and ending base of the fragment of 153 bp employed for cloning in the RNAi expression vector.

Response: Thank you very much for your suggestion. The fragment of 153- bp for construction of the RNAi expression vector was added in the materials and methods section (Lines 281 - 283 in Page 12).

Point 4:  In Figures 2, 5 and 11, it would be better to avoid the interruption of the Y axis, perhaps separating into two figures in the case of male and female inflorescences for Fig. 5 (the total has no meaning given the number of female inflorescences), and separating also JA and JA-Ile for Fig. 11.

Response: Thank you very much for your suggestion. The interruption of the Y axis was revised in Figures 2, 5 and 11, and the related composite histograms were separated in Figures 5 and 11. The histogram about the total flower number was discarded in Figure 5.

Reviewer 2 Report

The manuscript number ijms-994450 presents a very interesting finding, that JcDAD1 gene might have a differential function in flower development in Jatropha curcas contrast to what is already known for DAD1 in Arabidopsis. 

The novelty however of this finding is over missed while reading the manuscript. Therefore I suggest a revision to be conducted and especially the discussion part to be modified.

The discussion part is inadequate and needs to be re-written and re-structured. One major question that arises is why Jatropha plants react in JA so differently? Do the Jatropha have any other specific peculiarities? While reviewing their manuscript the authors need to answer these questions.

Author Response

Point 1: The manuscript number ijms-994450 presents a very interesting finding, that JcDAD1 gene might have a differential function in flower development in Jatropha curcas contrast to what is already known for DAD1 in Arabidopsis. The novelty however of this finding is over missed while reading the manuscript. Therefore I suggest a revision to be conducted and especially the discussion part to be modified. The discussion part is inadequate and needs to be re-written and re-structured.

Response: Thank you very much for your suggestion. Although we included as much discussion content as possible (Lines 218 ­ 219 and Lines 235 - 239), the discussion part was still inadequate. The new “Conclusions” section (Lines 307 - 318 in Page 13) was added to this article according to the template for articles in the International Journal of Molecular Sciences.

Point 2: In the discussion part, one major question that arises is why Jatropha plants react in JA so differently? Do the Jatropha have any other specific peculiarities?

Response: Thank you very much for your suggestion. Compared with that of Arabidopsis, silencing of JcDAD1 caused an increase in flower size and did not affect anther and pollen development in Jatropha. Previous studies showed that JA, a pivotal phytohormone, plays diversified roles in inflorescence and flower development in different species (e.g., Arabidopsis, rice and maize) (Reference 49 of the manuscript). Meanwhile, the lipase DAD1 also functions differently in various species (e.g., Arabidopsis and tomato) (References 22 and 48 of the manuscript). Thus, we speculate that the reason for the disparate reaction of Jatropha to JA is the diverse roles of JA in different species. In addition, we speculate that a threshold value of JA level may exist in JA regulation of Jatropha flower development because the lowest endogenous JA level in the transgenic JcDAD1-RNAi plants still accounts for approximately 26.76% of that in wild-type Jatropha plants.

Round 2

Reviewer 2 Report

I believa that the authors have succesfully adressed my previous comments. The manuscript can now be published.